# Tailoring the Electrical Energy Storage Capability of Dielectric Polymer Nanocomposites via Engineering of the Host–Guest Interface by Phosphonic Acids

**DOI:** 10.3390/molecules27217225

**Published:** 2022-10-25

**Authors:** Shaojing Wang, Peng Xu, Xiangyi Xu, Da Kang, Jie Chen, Zhe Li, Xingyi Huang

**Affiliations:** 1State Grid Shanghai Municipal Electric Power Company, Shanghai 200437, China; 2Shanghai Key Lab of Electrical Insulation and Thermal Aging, Shanghai Jiao Tong University, Shanghai 200240, China; 3Department of Electrical Engineering, Shanghai Jiao Tong University, Shanghai 200240, China

**Keywords:** phosphate-modified BaTiO_3_, nanocomposites, dielectric properties, breakdown strength, energy storage capability

## Abstract

Polymer nanocomposites have attracted broad attention in the area of dielectric and energy storage. However, the electrical and chemical performance mismatch between inorganic nanoparticles and polymer leads to interfacial incompatibility. In this study, phosphonic acid molecules with different functional ligands were introduced to the surface of BaTiO_3_ (BT) nanoparticles to tune their surface properties and tailor the host–guest interaction between BT and poly(vinylideneflyoride-*co*-hexafluroro propylene) (P(VDF-HFP)). The dielectric properties and electrical energy storage capability of the nanocomposites were recorded by broadband dielectric spectroscopy and electric displacement measurements, respectively. The influence of the ligand length and polarity on the dielectric properties and electrical energy storage of the nanocomposites was documented. The nanocomposite with 5 vol% 2,3,4,5,6-pentafluorobenzyl phosphonic acid (PFBPA)-modified BT had the highest energy density of 12.8 J cm^−3^ at 400 MV m^−1^, i.e., a 187% enhancement in the electrical energy storage capability over the pure P(VDF-HFP). This enhancement can be attributed to the strong electron-withdrawing effect of the pentafluorobenzyl group of PFBPA, which changed the electronic nature of the polymer–particle interface. On the other hand, PFBPA improves the compatibility of the host–guest interface in the nanocomposites and decreases the electrical mismatch of the interface. These results provide new insights into the design and preparation of high-performance dielectric nanocomposites.

## 1. Introduction

Polymer dielectrics are widely applied in high energy storage devices and organic field-effect transistors due to their advantages of low cost and easy processing [1,2,3,4]. However, most commercial polymers, with a rather low dielectric constant and energy storage density, cannot meet the requirements of modern energy storage devices [5,6,7]. Generally, the energy density (U) of linear dielectrics can be expressed as U = 1/2*kk*_0_*E*_b_^2^, where *k* is the dielectric constant, *E*_b_ is the maximum electric field that the dielectric can withstand without breakdown, and *k*_0_ is the vacuum permittivity [8,9]. Obviously, the dielectric constant and electric breakdown strength are two important parameters of the energy storage of dielectric materials.

Ceramic materials such as BaTiO_3_ (BT) have a high *k* value but generally suffer from a low breakdown strength and poor processability. Polymers usually have a high breakdown strength and easy processing but a low *k* value. A simple method was proposed to embed ceramic fillers as reinforcement materials into polymer matrices, utilizing the advantages of both the ceramics and the polymers [1,10,11,12,13]. However, there are still some challenges to overcome in achieving a high electrical energy density in polymer nanocomposites. For example, most nanoparticles have a high surface energy, which does not match the polymer matrix [14,15]. The presence of nanoparticle aggregation causes electric field distortion, resulting in a low breakdown strength of the nanocomposite films. In addition, the electrical migrate pathway provided by the nanoparticles in the composite results in an increase in the conduction loss. Recent studies have shown that there are three strategies for the preparation of nanocomposites with a high *k* value and high breakdown strength: (1) the construction of core-shell polymer@nanoparticle composite materials by in situ polymerization of monomers on nanoparticle surfaces [11,16], for example, wrapping the oxide film on the surface of Al particles, which can prevent electronic conduction between the fillers [14]; (2) building sandwich-structured composite materials [17,18], such as a boron nitride interlayer in a polymer film, which can inhibit electrical branches [8]; and (3) surface modification of ceramic nanoparticles for nanocomposites [19]. Among them, the surface modification technology provides a simple method for tuning the surface energy of nanoparticles. For example, it was reported that H_2_O_2_-modified BT nanoparticle surfaces were used to prepare PVDF-based composites. Because of the hydrogen bond interaction between the F atom and the hydroxyl group, the movement of the PVDF chain is restricted; thus, the nanocomposites have a low dielectric loss and a relatively stable high *k* value with increasing frequency and temperature [20]. The modification of the BT nanoparticle surfaces with molecules containing F generates a thin passivation layer with a high interfacial charge to slow down electrons moving towards them, thus enhancing the breakdown strength [21]. However, if the modification is incomplete, the residual surfactant may cause a high leakage current and increase the dielectric loss [6,7,22]. It was found that the formation of chemical bonds between the phosphonic acid molecules and BT nanoparticle surface can greatly reduce the surface energy of BT and improve the dispersion of BT in a polymer matrix [23]. Several types of phosphonic acid molecules were utilized as modifiers while only the modifier containing PO(OH)_2_ groups showed a high and robust interaction with BT nanoparticles, yielding coverage of the particles. Experimentally, phosphonic-acid-modified BT nanoparticles resulted in high *k* enhancement of the polymer (*k* ~ 37 at a BT nanoparticle loading of 50 vol%). Unfortunately, the electrical energy storage density was not efficiently increased because of the low dielectric strength [22,23,24,25]. In addition, the influence of the ligand length and polarity of the phosphonic acid molecules on the dielectric properties and electrical energy storage of nanocomposites was not documented in detail.

In this work, the influence of the ligand length and polarity of the phosphonic acid molecules on the dielectric and electrical energy storage properties of the poly(vinylideneflyoride-*co*-hexafluroro propylene) (P(VDF-HFP)) nanocomposites was investigated. BT nanoparticles were first modified with two types of phosphonic acid molecules with different functional ligands, including non-polar alkyl and polar aryl groups. Then, the resulting nanocomposites were prepared and characterized by broadband dielectric spectroscopy and electric displacement measurements. The results show that the nanocomposites comprising 2,3,4,5,6-pentafluorobenzyl phosphonic-acid-modified BT and P(VDF-HFP) possessed the highest dielectric constant, lowest dielectric loss, and highest discharged energy density. This strategy provides new insight into the fabrication of high-performance dielectric polymer nanocomposites via engineering of the surface of BT nanoparticles.

## 2. Results

### 2.1. Characterization of the Modified BT

Phosphonic acids containing six different functional groups were reacted with BT, which is expected to form a single layer of coverage on the surface of the BT nanoparticles (Figure 1). The TGA measurements (Appendix A) presented strong evidence for the successful preparation of modified BT. A significant weight loss of the modified BT (3.6–4.3% when heated above 550 °C) was observed in comparison with the pristine BT (1.1%). The calculated density of the phosphonic acid coverage was 4–17/nm^2^ provided that the density of BT was 6.08 g cm*^−^*^3^ and the BT nanoparticles were regarded as an ideal sphere with a diameter of 100 nm, which is greater than the theoretical monolayer value (a factor of 1.2–1.6/nm^2^) [26]. This is because most of the BT nanoparticles have a diameter of less than 100 nm.

A typical SEM image shown in Appendix A presents the morphology of the modified BT. In the figure, benefiting from the shielding shell, it is difficult to observe large aggregations of BT nanoparticles in a wide vision and the pristine BT nanoparticles mainly retain a nanoscale size. To investigate the effect of surface modification on the dielectric properties of BT, the nanoparticles were pressed into a compacted disk using KBr, which is widely used to prepare dick samples for IR spectroscopy measurements. Appendix A shows the frequency dependence of the dielectric properties of unmodified BT and modified BT from 0.1 Hz to 1 MHz at room temperature. One can observe that compared with the unmodified BT, the modified BT has a much lower dielectric constant and dielectric loss tangent. On the other hand, both the dielectric constant and dielectric loss tangent of the modified BT show a weak dependence on the frequency. The decrease in both the dielectric constant and dielectric loss tangent should originate from the introduction of the phosphonic acid monolayer, which not only removed the absorbed impurities (e.g., water, ions) but also changed the surface chemistry, namely, from hydroxyl-group-terminated surface to a phosphonic-acid-coated surface. In this case, the leakage currents and interfacial polarization can be significantly suppressed, resulting in the corresponding weak frequency dependence and lower values of both the dielectric constant and dielectric loss. All these characteristics suggest the successful preparation of modified BT.

### 2.2. Dielectric Properties of Nanocomposite Films

Electrical mismatch between polymer matrices and inorganic nanofillers can lead to severe electric field distortion and a significant decrease in the breakdown strength [27]. In this work, BT nanoparticles covered by phosphonic acid were facile prepared, and the aforementioned test results show that the dielectric constant and dielectric loss of the BT nanoparticles were simultaneously decreased. In this case, the electrical mismatch between P(VDF-HFP) and modified BT should be suppressed, which may lead to improved dielectric properties of the nanocomposite materials. Figure 2 shows the frequency dependence of the dielectric constant and dielectric loss of the nanocomposite films with different volume fractions of modified BT. As expected, the introduction of the BT nanoparticles results in dielectric enhancement of P(VDF-HFP) because of the high dielectric constant of BT. The dielectric constant of pristine P(VDF-HFP) is about 9.5 at 1 kHz [8] while PFBPA@BT/P(VDF-HFP) shows a high dielectric constant of 24 at 20 vol% nanoparticle loading.

In order to understand the influence of surface modification on the dielectric constant of P(VDF-HFP) nanocomposites in detail, the dielectric constant at 1000 Hz and the dielectric strength (dispersion) (∆*k* = *k_l_* − *k_h_*, where *k_l_* and *k_h_* are the low- (0.1 Hz) and high-frequency (1 × 10^7^ Hz) limits of the real dielectric constant, respectively) are summarized in Appendix A. One can observe that, for both types of modifiers, the dielectric constant of the nanocomposites exhibits the same tendency at each nanoparticle loading. In the case of alkyl phosphonic acids, the dielectric constant of the nanocomposites increases with the length of the terminal alkyl chains of the phosphonic acids. For the phenyl phosphonic acids, the dielectric constant of the nanocomposites increases with the length of the terminal alkyl chains of the phosphonic acids. However, there is no apparent difference between the effects of the two types of modifiers on the dielectric constant of the nanocomposites at a relatively low loading (i.e., ≤15 vol%). At a high loading of 20 vol%, the fluorobenzyl phosphonic acids result in a much higher dielectric constant in comparison with the others.

In the frequency range of 0.1 to 10^7^ Hz and at room temperature, the dielectric dispersion of the dielectric polymer nanocomposites should mainly originate from interfacial and dipolar polarization while electrical conduction may contribute to it. Here, as shown in Appendix A, PFBPA@BT/P(VDF-HFP) always shows the highest dielectric dispersion among the nanocomposites at each loading. However, at nanoparticle loadings lower than 10 vol%, all nanocomposites show low and comparable dielectric dispersion apart from PFBPA@BT/P(VDF-HFP). Starting from 20 vol%, the phenyl-phosphonic-acid-modified BT shows higher dielectric dispersion of the nanocomposites in comparison with the alkyl-phosphonic-acid-modified BT, although the longer alkyl chains of phosphonic acids tend to dielectric dispersion of the nanocomposites when the BT loading increases.

In the case of dielectric loss, as shown in Figure 3, the main difference appears at low frequencies. At room temperature, the dielectric relaxation (loss) of hot-pressed PVDF in high (about 10^6^ to 10^7^ Hz) and low frequencies (about 10 Hz) mainly originates from the glass transition (β-relaxation, i.e., rearrangement of the segmental dipole of amorphous regions) and αc-relaxation (i.e., molecular motions within the crystalline phase), respectively [1,11].

Figure 3 suggests that the nanocomposites show a high dielectric loss both in low-frequency and high-frequency regions. For the P(VDF-HFP)-based nanocomposites, the low-frequency and high-frequency dielectric losses are generated from the electrical conduction process, interfacial polarization, and amorphous segmental motion, respectively. In the low-frequency region (0.1–100 Hz), as the modified BT increases from 5% to 15 vol%, the dielectric loss of the nanocomposite films first decreases and then increases with the modified BT increase to 20 vol% [24]. This is because in the low-frequency region, electrical conduction and interfacial polarization play a dominant role in dielectric loss and the excess of BT increases the leakage current, leading to a high dielectric loss. In the high-frequency region (above 10^5^ Hz), the dielectric losses of these nanocomposite films are slightly reduced or remain at the same level as the volume fractions of modified BT increase, and the organic phosphonic acid monolayer shield shell of the BT nanoparticles shows a strong interaction with the P(VDF-HFP) matrix, which limits the mobility of the macromolecular chains, leading to a low dielectric loss.

The temperature-dependent dielectric spectra of the proposed nanocomposite films were further investigated from 0 to 140 °C as shown in Figure 4. It can be observed that as the temperature increases, the dielectric constant of the nanocomposite films increases. This phenomenon can be attributed to the increasing movement of chain segments of P(VDF-HFP), which results in a higher dielectric constant. However, the relationship between the dielectric loss and temperature is more complicated. For instance, all the dielectric loss curves in Figure 4 show a loss peak between −40 and −20 °C, which is closely related to T_g_ of P(VDF-HFP) [28,29]. Another striking feature of these nanocomposite films is the temperature sensitivity of the dielectric properties. It is noted that the nanocomposite films with phenyl-containing BT have a higher temperature sensitivity than those with alkyl-containing BT. On the one hand, the dielectric loss of the former remarkably increases at 50 °C while the latter is around 100 °C. On the other hand, the dielectric constants of BPA@BT/P(VDF-HFP), FPMPA@BT/P(VDF-HFP), and PFBPA@BT/P(VDF-HFP) increase by 219%, 206%, and 335% as the temperature increases from −50 to 140 °C, respectively; while, the dielectric constants of HPA@BT/P(VDF-HFP), NOPA@BT/P(VDF-HFP), and ODPA@BT/P(VDF-HFP) only increases by 180%, 210%, and 166%, respectively. This phenomenon can be explained by the existence of the electron-withdrawing F atom changing the polarity of the nanoparticle surface, enhancing the interfacial polarization and resulting in a high dielectric constant. However, the dielectric constant of the nanocomposite films is opposite to that of the modified BT, indicating that the BT particles containing the long alkyl chain or fluorinated aromatic rings have a better incorporation with P(VDF-HFP) [27].

To further understand the effect of the molecular interaction between the fillers and the polymer matrix at the interface, the temperature- and frequency-dependent imaginary parts of the electrical modulus (M″) are provided in Figure 5a. Two peaks can clearly be observed, which are assigned to the relaxation of chain segments in the amorphous phase (in the low-temperature and high-frequency regions) and interfacial polarization (in the high-temperature and low-frequency range), respectively. Obviously, under the same filler loading, the interfacial polarization peak of FPMPA@BT/P(VDF-HFP) is weaker than that of the other five nanocomposites, demonstrating that the strong molecular interaction significantly improves the compatibility of the two components and effectively suppresses the high polarization hysteresis loss caused by interface polarization. The activation energy (*E_a_*) in the interfacial area can be calculated by the following Arrhenius plot [4,18]:(1)$$\ln f_{\max}=\ln f_{0}-\frac{E_{a}}{KT}$$
where *f*_max_ denotes the peak frequency of M″ under a certain temperature (*T*), *f*_0_ is the pre-exponential factor, and *K* is the Boltzmann constant. As shown in Figure 5b, *E_a_* of FPMPA@BT/P(VDF-HFP) is the highest among the six nanocomposites, indicating that the molecular interaction can increase the energy of the space charge migration from the interface to other regions, thereby enhancing the dielectric strength.

### 2.3. Breakdown Electric Field

Figure 6 shows the Weibull plots for the breakdown strength of the six types of nanocomposite films, which all exhibit a decreased breakdown strength with the increase in the BT loading. This is consistent with the theoretical calculation [29]. The decreased breakdown strength is mainly attributed to the large electrical mismatch between BT and P(VDF-HFP), which results in large electric field distortion in the filler/matrix interface. It was also found that in the nanocomposites, the breakdown strength is closely associated with the surface chemistry of the BT nanoparticles. At a given BT loading level, the nanocomposite films with aryl-group-functionalized BT nanoparticles have a much higher breakdown strength in comparison with those with alkyl-group-functionalized nanoparticles. In the case of alkyl group functionalization, the longer alkyl chain results in a higher breakdown strength of the nanocomposites. For the aryl group functionalization, the fluorinated aromatic rings tend to result in a higher breakdown strength in the nanocomposites. In addition, the breakdown strength of PFBPA@BT/P(VDF-HFP) is higher in comparison with FPBPA@BT/P(VDF-HFP). For all the nanocomposites, PFBPA@BT/P(VDF-HFP) has the highest breakdown strength at a given BT loading level.

β represents the scatter of the breakdown strength data and can be calculated from the slope of the linear fitting curves. In the case of alkyl group functionalization, the β values of the nanocomposites are mainly in the range of 10 to 20, and there is no correlation between the BT loading/surface chemistry and the β values. For the aryl group functionalization, the β values of nonfluorinated aryl (i.e., BPA) functionalized BT nanocomposites are also low and there is no correlation between the BPA@BT loading and the β values. However, in the case of the fluorinated aryl-functionalized BT, the nanocomposites exhibit much higher β values and β increases with the BT loading level. Among the nanocomposites, PFBPA@BT/P(VDF-HFP) shows the highest β value of 174.7 at 20 vol% BT loading, suggesting a highly homogeneous microstructure of the nanocomposites.

### 2.4. Electrical Energy Storage

In addition to the energy density, the charge-discharge efficiency (η) is another important parameter for dielectric materials, which is calculated by the following equation: U=∫EdD [30], where E and D are the electric field and electric displacement, respectively. The total energy density and discharged energy density of nanocomposite films were calculated by the D–E loop curves. Figure 7 displays the total energy density and discharged energy density of HPA@BT/P(VDF-HFP), NOPA@BT/P(VDF-HFP), ODPA@BT/P(VDF-HFP), BPA@BT/P(VDF-HFP), FPMPA@BT/P(VDF-HFP), and PFBPA@BT/P(VDF-HFP) with different volume fractions of modified BT, respectively. It can be seen that as the content of modified BT increases from 5 to 20 vol%, the stored energy density of the six samples does not exhibit much improvement because of the inferior breakdown strength, which may be attributed to the electrical conduction nature of BT [31,32,33]. For instance, *U_s_* and *U_r_* of 5 vol%-HPA@BT/P(VDF-HFP) are 4.4 and 2.5 J cm^−3^ and the calculated η is 57%. When the volume fraction of the modified BT increases to 20 vol%, *U_s_* and *U_r_* decrease to 3.8 and 2.0 J cm*^−^*^3^. Notably, under the four different contents of modified BT, nanocomposite PFBPA@BT/P(VDF-HFP) exhibits a higher *U_s_* and *U_r_* than the other five nanocomposites. For example, under 5 vol%, PFBPA@BT/P(VDF-HFP) exhibits *U_s_* of about 12.8 J cm^−3^ and *U_r_* of 7.7 J cm^−3^, which might be ascribed to the suppressed dielectric loss and superior breakdown strength as discussed before. With an increase in the content of PFBPA@BT, *U_s_* of the nanocomposite shows a slight vibration but *U_r_* a gradually decreases 4.3 J cm^−3^ when the volume fraction reaches 20%. This is mainly ascribed to the decrease in η from 60% to 33%. These results suggest that nanocomposites with a high content of BT are not suitable for high-performance dielectric materials. In addition, PFBPA is more effective in improving the energy storage performance of nanocomposites than the other five phosphonic acids.

## 3. Materials and Methods

### 3.1. Materials

Commercial tetragonal-phase BT particles with an average diameter of 100 nm were purchased from Shandong Sinocera Functional Material Company, China. Benzylphosphonic acid (BPA), [(4-fluorophenyl)methyl]-phosphonic acid (FPMPA), 2,3,4,5,6-pentafluorobenzyl phosphonic acid (PFBPA), *n*-octylphosphonic acid (NOPA), hexylphosphonic acid (HPA), and octadecylphosphonic acid (ODPA) were purchased from Sigma-Aldrich. P(VDF-HFP) was supplied by SOLVAY. DMF was purchased from Sinopharm Chemical Reagent. Ethanol was bought from Tansoole (China) and used as received.

### 3.2. Methods

Thermogravimetric analysis (TGA) was carried out by a NETZSCH TG209 F3 (Germany) instrument with a heating rate of 20 °C min^−1^ from 30 to 800 °C under nitrogen flow. The morphology of the modified BT was characterized by a Nova NanoSEM scanning electron microscope (SEM, PEI Company, Boulder, CO, USA). Firstly, the modified BT samples were dispersed in ethanol by ultrasonication. Then, the mixtures were cast onto silicon plates and dried in an oven at 40 °C for 8 h. Finally, all the samples were recorded by SEM before being sputter-coated with a gold layer to avoid the accumulation of charges. The dielectric properties of the modified BT and nanocomposite films were measured using a Novocontrol Alpha-N high-resolution dielectric analyzer (GmbH Concept 40) from 10^−1^ to 10^6^ Hz at various temperatures (−50 to 140 °C). Gold electrodes were evaporated on the front and rear surfaces of the samples. The breakdown strength of the nanocomposite films was recorded by a DC voltage strength tester (Shanghai Lanpotronics Co., Shanghai, China) under a rate of 200 V s^−1^. Every sample was tested with 12 different plots for the Weibull statistical distribution analysis. The Weibull statistical distribution in the case of the ramp voltage test can be written as following equation:$$P\left(E\right)=1-\exp\left[-{\left(\frac{E}{E_{0}}\right)}^{\beta }\right]$$
where *E* is the experimental breakdown strength; *P* is the cumulative probability of electrical failure; *β* is the shape parameter, which is related to the scatter of the data; and *E*_0_ is the characteristic breakdown strength, which represents the breakdown strength at the cumulative failure probability of 63.2%, which is often used to compare the breakdown strength of various samples. Commonly, this parameter is used to compare differences in the breakdown strength among specimens. A simpler approximation for the most likely probability of failure is recommended by the IEEE 930-2004 standard as the following equation:$$P_{i}=\frac{i-0.44}{n+0.25}\times 100\%$$
where *i* is the *i*-th result when the values of *E* are sorted in ascending order and *n* is the corresponding number of samples; in our study, *n* = 12.

The electrical energy storage capability of the nanocomposite films was evaluated by electric displacement–electric field (*D*–*E*) loops, which were recorded using a Precision Premier II ferroelectric polarization tester (Radiant, Inc.) at room temperature. The thickness of the nanocomposite films was around 15 μm. The energy densities (*U*_e_) were extracted from the *D*–*E* loops based on the following equation:$$U_{\theta }=\int EdD$$
where *E* and *D* are the electric field and electric displacement, respectively.

Based on the results of the *D*–*E* loops, the charge-discharge efficiency (*η*) was calculated by the following equation:$$\eta =\frac{U_{r}}{U_{s}}=1-\frac{U_{l}}{U_{s}}$$
where *U_s_* is the stored energy density, *U_r_* is the released energy density, and *U_l_* is the loss energy density, respectively.

## 4. Conclusions

We have prepared a series of phosphonic-acid-modified BT nanoparticles, which were used to prepare P(VDF-HFP)-based nanocomposite films. The surface properties of the BT nanoparticles and the interfacial adhesion between the nanoparticles and polymer matrix were improved by tuning the surface energy of the BT nanoparticles via different phosphonic acid pendants. It was found that the polarity ligands of the phosphonic acids played a more significant role in enhancing the dielectric properties and energy storage capacity. The dielectric constant of P(VDF-HFP) was significantly enhanced after the introduction of modified BT nanoparticles. Specifically, 20 vol%-PFBPA@BT/P(VDF-HFP) showed the highest dielectric constant of about 24, which is about 3 times higher than pristine P(VDF-HFP). Meanwhile, the nanocomposite with PFBPA exhibited a superior energy storage performance compared to the other five nanocomposites. For instance, 5 vol%-PFBPA@BT/P(VDF-HFP) exhibited *U_e_* of 7.7 J cm^−3^ while *U_e_* of the other five nanocomposites was lower than 5 J cm^−3^. These results indicate that among the six phosphonic acids, PFBPA is more suitable for the modification of BT to fabricate high-performance energy storage nanocomposites.

## Figures and Tables

**Figure 1 molecules-27-07225-f001:**
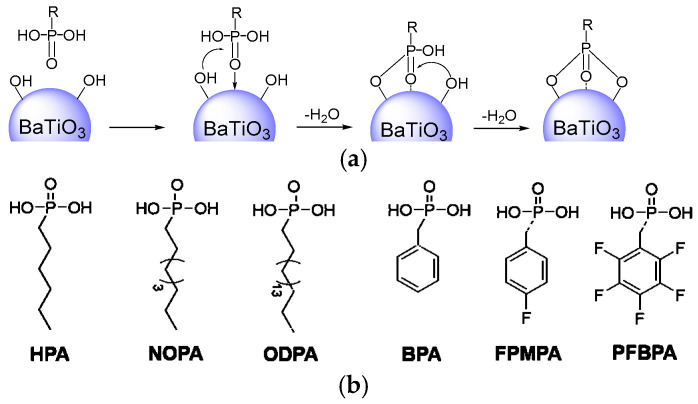
Mechanism of phosphonic acid binding to the surface of BT nanoparticles and the electrostatic potential maps of phosphonic acid modifiers obtained from DFT modeling. The red color indicates negative charge, whereas the blue color indicates positive charge. (**a**) Binding mechanism between phosphonic acid and BT. (**b**) Electrostatic potential maps of phosphonic acids.

**Figure 2 molecules-27-07225-f002:**
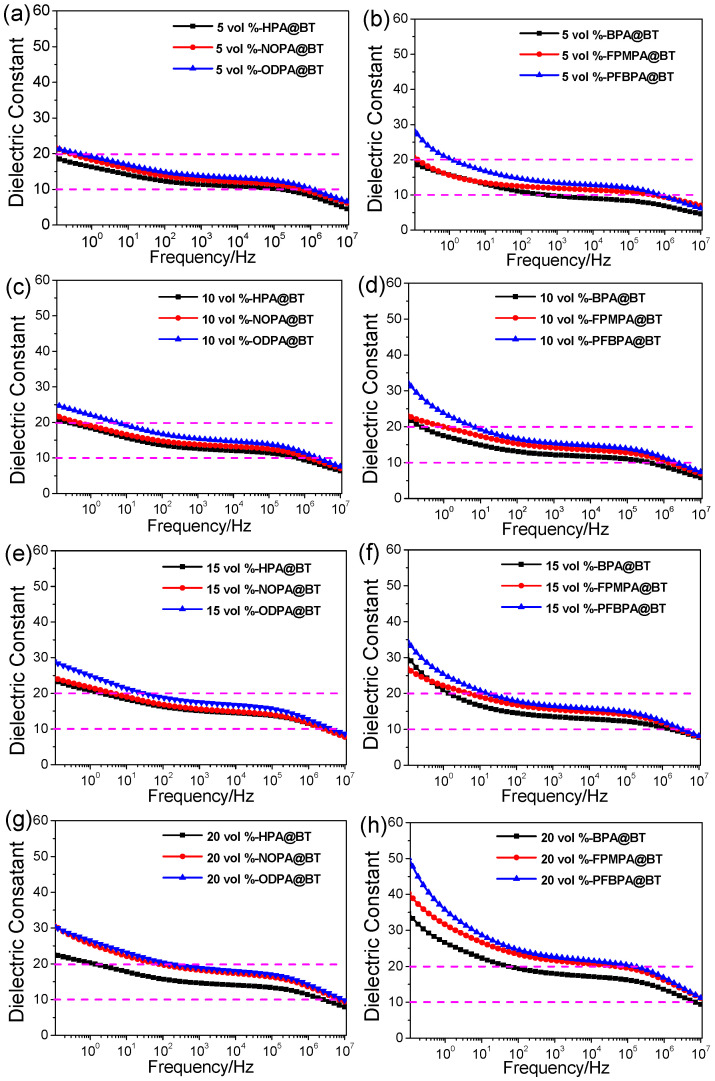
Frequency dependence of the dielectric constant of the P(VDF-HFP) nanocomposite films with different volume fractions of modified BT (**a**–**h**).

**Figure 3 molecules-27-07225-f003:**
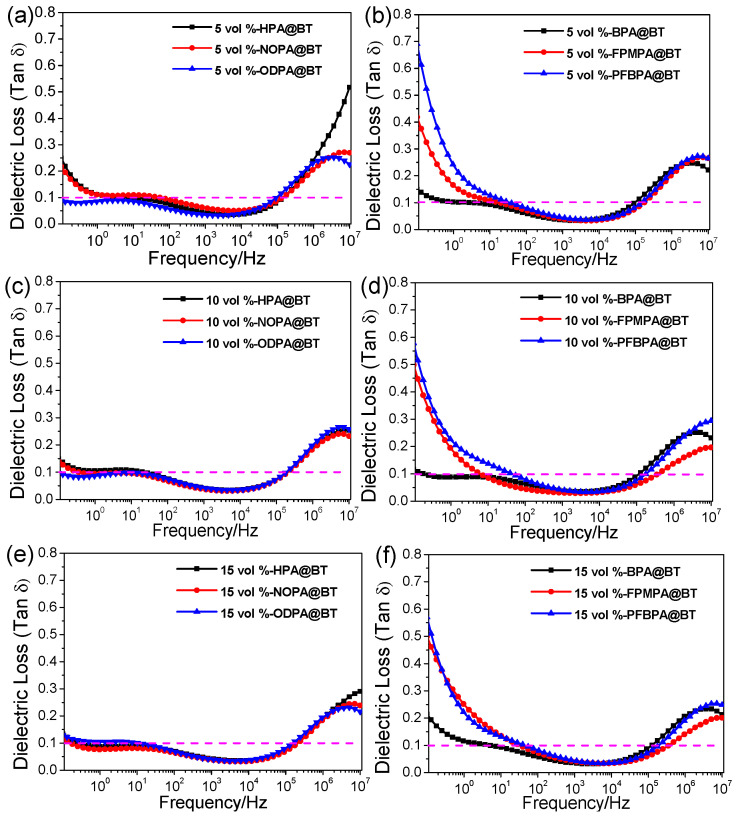
Frequency dependence of the dielectric loss of the P(VDF-HFP) nanocomposite films with different volume fractions of modified BT (**a**–**h**).

**Figure 4 molecules-27-07225-f004:**
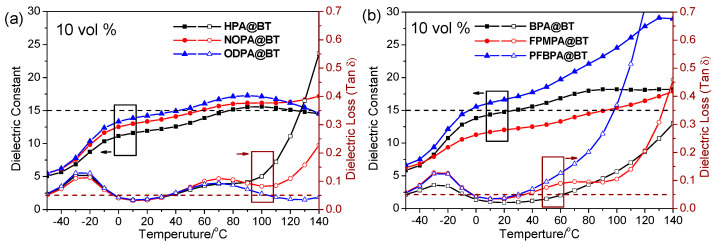
Temperature-dependent dielectric spectra of representative 10 vol% loading nanocomposite films (**a**,**b**).

**Figure 5 molecules-27-07225-f005:**
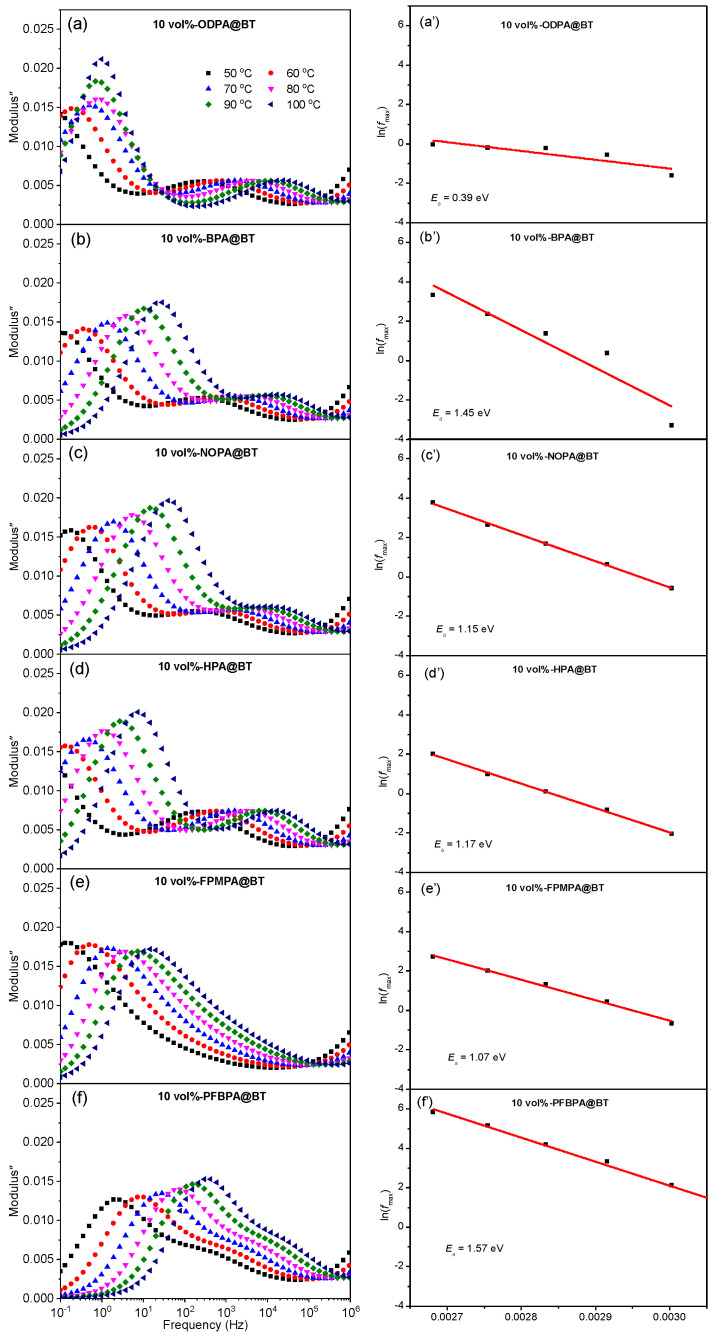
Frequency dependence of the imaginary electric modulus (**a**–**f**) and 1/T Arrhenius plots (**a′**–**f′**) of the P(VDF-HFP)-based nanocomposites with 10 vol% of modified BT nanoparticles.

**Figure 6 molecules-27-07225-f006:**
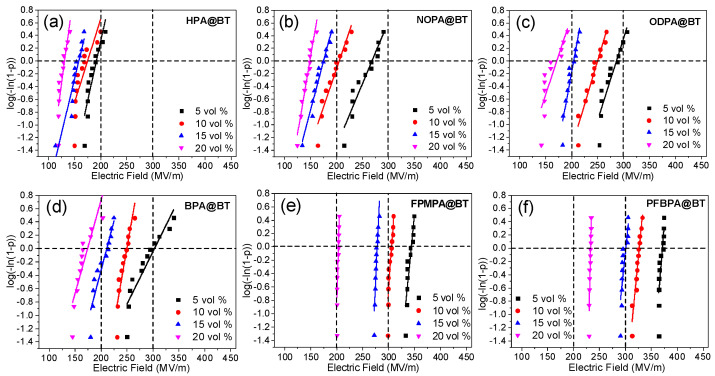
Weibull plots for the breakdown strength of the P(VDF-HFP) nanocomposites with different volume fractions of functionalized BT nanoparticles (**a**–**f**).

**Figure 7 molecules-27-07225-f007:**
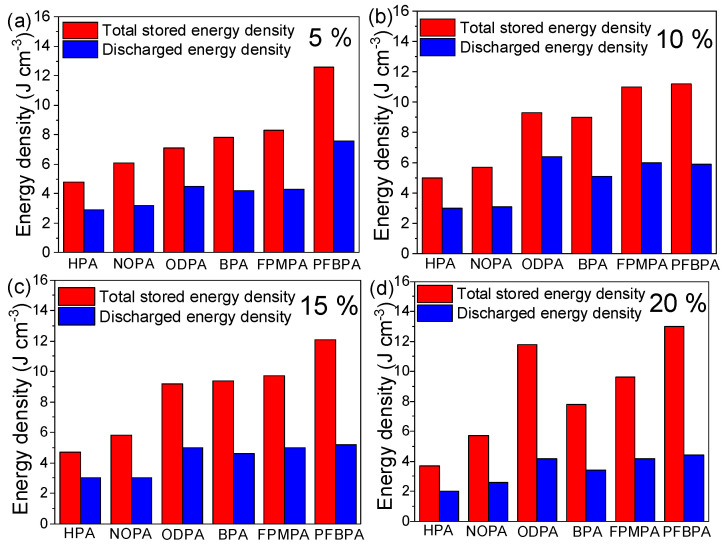
Total energy density and discharged energy density of nanocomposite films with different volume fractions of modified BTs: (**a**) 5 vol%, (**b**) 10 vol%, (**c**) 15 vol% and (**d**) 20 vol%.

## Data Availability

The data presented in this study are available upon request from the corresponding author.

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
