# Peer review of "Tailoring the Electrical Energy Storage Capability of Dielectric Polymer Nanocomposites via Engineering of the Host–Guest Interface by Phosphonic Acids"

_molecules, 2022, doi:10.3390/molecules27217225_

Round 1

Reviewer 1 Report

The paper concerns with a very important topic in the materials science field in the area of dielectric and energy storage. The experimental plot sounds good and the paper is rich of data and analyses. The paper merits the pubblication. Nevertheless some drawbacks must be resolved.

- At first a deep revision of language and sintax is needed. A lot of mistakes/misprints (even in the subtitles, for instance in the abstract - lines 12-13 - the verb is missing or at Subsection 2.3 FIELD instead of FILED) lead to a difficult reading of the paper.
- Section 3 "DIscussion" is very short and the text do not discuss the results but only summarize them, resembling a conclusion section. The authors should change the section titles and rearrange the distribution. - Lines 190-191 The sentence seems to be a little tautological concerning the role of temperature. - The definition of efficiency refers to ref 20 but is better to cite other papers, for instance  ACS Appl. Mater. Interfaces 2019, 11, 36824−36830 or similar.  - The authors must add some comments about the role of polar BTO nanoparticles trying to comment about the interactions between different polar "clusters" (organic vs inorganic).   

Author Response

For Reviewer 1:

The paper concerns with a very important topic in the materials science field in the area of dielectric and energy storage. The experimental plot sounds good and the paper is rich of data and analyses. The paper merits the pubblication. Nevertheless some drawbacks must be resolved.

Thanks for your kind comments. According to your nice advices, here we answered the above questions one by one after careful consideration.

(1) At first a deep revision of language and sintax is needed. A lot of mistakes/misprints (even in the subtitles, for instance in the abstract - lines 12-13 - the verb is missing or at Subsection 2.3 FIELD instead of FILED) lead to a difficult reading of the paper.

Response: Very sorry for these errors, the language in the manuscript has been polished to improve readability.

(2) Section 3 "Discussion" is very short and the text do not discuss the results but only summarize them, resembling a conclusion section. The authors should change the section titles and rearrange the distribution.

Response: The title of Section 3, "Discussion", is replaced with "Conclusion"

(3) Lines 190-191 The sentence seems to be a little tautological concerning the role of temperature. - The definition of efficiency refers to ref 20 but is better to cite other papers, for instance  ACS Appl. Mater. Interfaces 2019, 11, 3682436830 or similar. 

Response: This sentence in lines 190-191 has been modified. Ref. 20 with substitution as ACS Appl. Mater. Interfaces 2019, 11, 3682436830.

(4) The authors must add some comments about the role of polar BTO nanoparticles trying to comment about the interactions between different polar "clusters" (organic vs inorganic).   

Response: Thanks for your kind advice, and the examples of the role of polar BTO nanoparticles in a dielectric polymer have been added to the manuscript.

Reviewer 2 Report

Dear Editor/Author

The manuscript represents some quality data and novelty which may of interest to the readers of the journal. However the manuscript can be accepted for publication provided the authors attend the folowing queries.

1. The Introduction section is weak, needs significant improvement with latest references.

2. Number of references in the manuscript are inadequate, strongly recommend the authors to improve number of references and also, use the latest references (prefereably last 5 to 6 years)

3. Quality of SEM image used looks poor, No particle distributions are seen, what happned to BT particles in the composite? why they are not significantly apparent? must replave the SEM image with better quality to show the distribution of BT particles in the composite.

4. MUST Provide the C-V curves and Nyquist plots for BT and BT composite, else the energy storage claims are inappropriate.

5. Include the following reference in introdution section to show the advancement of research in PVDF based inorganic oxide nanocomposites

Journal of Materials Science: Materials in Electronics (2018) 29:10593–10599

Author Response

For Reviewer2:

The manuscript represents some quality data and novelty which may of interest to the readers of the journal. However, the manuscript can be accepted for publication provided the authors attend the folowing queries.

We thank the reviewer for these positive comments. For the above kind comments, we revised the manuscript one by one.

(1) The Introduction section is weak, needs significant improvement with latest references.

Response: The introduction section has been revised and updated with the latest literature.

(2) Number of references in the manuscript are inadequate, strongly recommend the authors to improve number of references and also, use the latest references (prefereably last 5 to 6 years)

Response: The references for almost five years have been added to the manuscript.

(3) Quality of SEM image used looks poor, No particle distributions are seen, what happned to BT particles in the composite? why they are not significantly apparent? must replave the SEM image with better quality to show the distribution of BT particles in the composite.

Response: We replaced the high-quality SEM images in the manuscript. Besides, SEM image of the distribution of BT particles in the polymer matrix have been added to the manuscript.

(4) MUST Provide the C-V curves and Nyquist plots for BT and BT composite, else the energy storage claims are inappropriate.

Response: We thank the referee for the valuable suggestion, as shown in Figure 3-8, the P-E curves of the BT composites have been added to the manuscript.

 (5) Include the following reference in introdution section to show the advancement of research in PVDF based inorganic oxide nanocomposites

Journal of Materials Science: Materials in Electronics (2018) 29:10593–10599

Response: A description of the research progress in PVDF-based inorganic oxide nanocomposites has been added to the manuscript and is cited in Journal of Materials Science: Materials in Electronics (2018) 29:10593–10599.
